# Personal Values, Attitudes and Travel Intentions Towards Cycling and Walking, and Actual Behavior

**Jesús García [1,\*], Lidón Mars [2], Rosa Arroyo [1], Daniel Casquero [1], Floridea di Ciommo [3] and Tomás Ruiz [1]**

1   Transport Department, School of Civil Engineering, Universitat Politécnica de Valéncia, 46022 Valéncia, Spain
2   Faculty of Health Sciences, Universidad Europea de Valencia, 46010 Valéncia, Spain
3   cambiaMO | cambia Movilidad, 28012 Madrid, Spain
*   Correspondence: jegarqui@cam.upv.es

**Abstract:** Personal values are psychological factors scarcely presented in travel behavior studies, despite their importance in determining life choices, decisions and actions. This paper contributes to filling this gap. The purpose of this study is to analyze the influence of personal values on attitudes, intentions and current cycling and walking. Data regarding personal values, attitudes, intentions and current use of cycling and walking were collected through a web-based survey. Pearson correlations, independent sample non-parametric tests and hierarchical regressions have been used to analyze the information. Reasonably weak but statistically significant direct and indirect effects have been identified between personal values and attitudes towards cycling and walking, intentions and current walking and cycling. Openness to change and self-transcendence values are associated to cycling and walking. Actions to encourage those personal values are needed to improve sustainable transport and mobility.

**Keywords:** travel behavior; personal values; cycling and walking; sustainable travel

## 1. Introduction

Psychological factors play an important role as explanatory variables of travel behavior. In particular, they are extremely useful to understand how travelers react to the implementation of both hard (e.g., car-use restrictions) and soft (e.g., awareness campaigns) transportation measures [1–4].

Personal values determine life choices, decisions and actions [5]. However, personal values have been very scarcely included in travel behavior studies of the transport planning field because results of empirical research indicate that the direct association between values and behavior [6–11] is weak and moderated by several factors [12–14], and they are relatively stable over time thus having a rather small importance in predictive models. However, the sustainable paradigm, the recent economic recession and the new urban mobility solutions are testing this aforementioned hypothesis. Therefore, it is important to identify to what extent personal values influence travel behavior and other psychological variables that are more frequently used in this area. The present study attempts to fill this important gap in the travel behavior and transport planning literature.

It is well known that attitudes towards travel modes and intentions to use each travel mode significantly influence travel behavior [15–22]. We assume that personal values, which are the basis of all human behavior, also play a meaningful role explaining attitudes, intentions and travel behavior.

The purpose of this work is to analyze the role of personal values in the study of attitudes, intentions and travel behavior focused on sustainable travel modes: Cycling and walking.

The current state of the research field is presented below. First, research findings of the direct influence of personal values on general travel behavior are revised. Then, the literature review

is focused on how personal values affect attitudes towards travel modes and intentions to travel. The conceptual framework and hypotheses end this section. Procedures used to obtain the data, a description of the participants and measures used to collect the information are presented in Section 2. The results of the data analysis using Pearson correlations, Mann–Whitney tests and hierarchical regressions are presented in Section 3. The paper ends with discussion and conclusions in Section 4.

## 1.1. Personal Values and Travel Behavior

Personal values are evaluations of abstract ideas (e.g., equality or honesty) in terms of their importance as guiding principles in people's life [23,24]. According to Schwartz [24], typology of values distinguishes between three universal requirements of human existence: Needs of individuals as biological organisms, requisites of coordinated social interactions and survival needs of groups. According to this author, values express ten types of motivation: Achievement, benevolence, conformity, hedonism, power, security, self-direction, stimulation, tradition and universalism. These values can be plotted in a circumflex structure to form four higher order value domains: Conservation, openness to change, self-enhancement and self-transcendence. These four higher order domains, in turn, represent two basic dimensions of value conflict. One dimension contrasts conservation values (e.g., national security) against openness values (e.g., freedom), whereas the other dimension contrasts self-enhancement (e.g., power) against self-transcendence values (e.g., helpfulness) (Schwartz [24]). We adopt the classification of values by Schwartz because it is the most used in the scarce literature on travel behavior in transport planning (Table 1).

**Table 1.** Contributions of literature on travel behavior in transport planning.

| Reference | Values Scale | Contribution |
|---|---|---|
| Hunecke et al. [14] | Schwartz scale | Self-enhancement negatively influenced bike use Openness to change positively affected the use of public transport |
| Lind et al. [15] | Value orientation measure | Awareness of consequences was positively associated to be in the active travelers. Ascription of responsibility was negatively associated to be in the active travelers |
| Nordlund and Garvill [25] | Schwartz scale | Self-transcendence and ecocentrism directly influence car use reduction |
| Nordlund and Westin [26] | Schwartz scale | Openness to change directly influenced on the intention to travel by a new railway line under construction. |
| Paulsen et al. [27] | Schwartz scale | Hedonism, security and power personal values influence attitudes towards car ownership |
| Pojani et al. [28] | Lifestyle orientations | Equality and materialism indirectly and positively influence the intention to use cars. The former indirectly and negatively influence the intention to use the bus. Equality was indirectly and positively related to the intention to cycle, but materialism directly and negatively affected the intention to cycle. |

According to the findings in the literature, the effects of values on behaviors are occasionally weak. Even then, several authors have studied these relations and identified some interesting results. People who score higher in idealism (extent to which self-related ideals are valued and used to determine behavior) should exhibit more value-behavior consistency [29]. Low self-monitors base their behaviors on their internal ideologies, values and attitudes, whereas high self-monitors base their behaviors on external, social cues. Low self-monitors may also be more likely to see their values as being relevant to their attitudes and behavior [30,31].

Some situational factors can decrease the impact of values by reducing people's capacity to express their values in their behavior. For example, time pressure might cause people to temporarily ignore an

important value, thereby reducing its impact on behavior. However, in general behavior should be more congruent with a value when the value is made salient than when it is not made salient [31–33].

In the transport planning field, Hunecke et al. [34] analyzed how psychological variables improved the analysis of travel mode choice. With respect to personal values, they found that self-enhancement negatively influenced bike use, whereas openness to change positively affected the use of public transport. Lind et al. [35] used cluster analysis and hierarchical logistic regressions to differentiate car versus active travelers. They used the value orientation measure [36,37], and found that awareness of consequences was positively associated to be in the active travelers' group. In contrast, ascription of responsibility was negatively associated to be in the active travelers' group.

## 1.2. Personal Values, Attitudes and Intention to Travel

Behavioral intentions were first introduced by Fishbein and Ajzen [38] in their Theory of Reasoned Action (TRA), which aims to measure behavioral intention as prediction of actual behavior. They described that intentions are "assumed to capture motivational factors that influence a behavior" and can also be a measure of how much effort someone is willing to exert when performing a behavior.

On the other hand, an attitude is "a relatively enduring organization of beliefs, feelings, and behavioral tendencies towards socially significant objects, groups, events or symbols" [39]. It denotes a psychological tendency that is expressed by evaluating a particular entity with some degree of favor or disfavor [40].

The TRA, and its extention, the Theory of Planned Behavior [41,42], have supported most of the existing empirical research in the travel behavior field regarding the influence of attitudes towards travel modes, and intentions to use travel modes, on travel behavior.

Some studies have found significant direct relationships between attitudes towards characteristics of travel modes and travel behavior [16,43–46]. But a larger amount of works have confirmed the attitude–intention–behavior framework, in which significant effects found between intentions to use travel modes and real use are stronger, and attitudes towards travel modes also significantly and directly influence to a greater extent both intentions and behavior [15,17–22,27,28,47–60].

A few works have studied the influence of personal values on travel attitudes and intentions to travel. Nordlund and Garvill [25] studied willingness to reduce car use, and found that self-transcendence and ecocentrism (an environmental value measured with Thomson and Bartons's scale [61]) directly influence problem awareness concerning biosphere and humankind, and personal norms concerning car use reduction. Nordlund and Westin [26] found that openness to change vs. conservation, and self-transcendence vs. self-enhancement, directly influence environmental concerns. They also found a direct influence of openness to change vs. conservation on the intention to travel by a new railway line under construction.

Paulsen et al. [27] adopted the value–attitude–behavior hierarchy proposed by Homer and Kahle [62] to study travel mode choice. Using an integrated choice and latent variable model, they confirmed Homer and Kahle's framework, and found that hedonism, security and power personal values influence attitudes towards flexibility, convenience and comfort, and car ownership.

Pojani et al. [28] studied the intentions to use car, bus and bicycle using different attitudes towards each travel mode, personal norms about the environment and lifestyle orientations that include some personal values. Lifestyle orientations towards equality and materialism were found to indirectly and positively influence the intention to use cars. The former indirectly and negatively influence the intention to use the bus. With respect to bike use, equality was indirectly and positively related to the intention to cycle, but materialism directly and negatively affected the intention to cycle.

## 1.3. Conceptual Framework and Hypotheses

Considering the evidences found in the literature, in this study the value–attitude–behavior hierarchy proposed by Homer and Kahle [62], and the Theory of Reasoned Action (Fishbein and

Ajzen [38]) are considered. Therefore, we assume a value–attitude–intention–behavior hierarchy as conceptual framework (Figure 1).

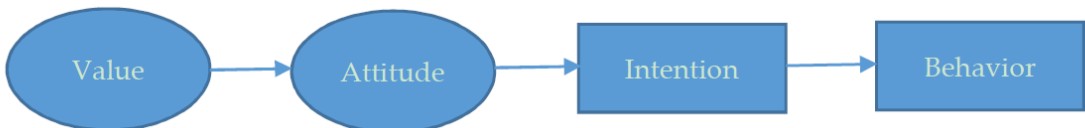

**Figure 1.** Conceptual framework.

The hypotheses defined below are focused on the relations between personal values and the other variables considered: Attitudes towards cycling and walking, intentions to cycle and walk, and current cycling and walking. The associations between the latter variables are not considered in detail, since they have been studied extensively elsewhere.

Directly derived from the conceptual framework adopted, we postulate direct effects of personal values on attitudes towards cycling and walking. But the influence of values on intentions and current cycling and walking is indirect.

**Hypothesis 1.** *Personal values will have a direct and significant effect on attitudes towards cycling and walking. The influence of values on intentions to cycle and walk will be indirect through attitudes. Similarly, the influence of values on current cycling and walking will be also indirect through attitudes and intentions.*

Bearing in mind the scarce literature on values and travel behavior, and how personal values are defined by Schwartz, we establish that openness to change and self-transcendence values (e.g., freedom or helpfulness) are positively associated to cycling and walking. On the other hand, conservation and self-enhancement values (e.g., national security or power) are negatively associated to cycling and walking. Translating these assumptions to the ten personal values of Schwartz, we have:

**Hypothesis 2.** *Benevolence, universalism, self-direction and stimulation will be positively associated with attitudes, intentions and current cycling and walking*; and

**Hypothesis 3.** *Hedonism, achievement, power, security, conformity and tradition will be negatively associated with attitudes, intentions and current cycling and walking.*

## 2. Methods

### 2.1. Procedure

The data collection effort was carried out using a web-based survey that was developed as part of the MINERVA (Estudio de la Movilidad de Personas Mediante Métodos Innovadores de Recogida de Datos) project, funded by the Ministry of Economy and Competitiveness of Spain. The survey was distributed online thanks to the contribution of several organizations who collaborated in the dissemination process, including universities, private companies and the regional government, among others. The data collection step took place in Valencia (Spain) as the main area of the study, although different locations were also accepted. The survey was open between May and October 2017, excluding August due to the vacation period in Spain which could have influenced negatively on the response rate. This long period allowed sending reminders and follow-ups to increase the participation in the survey.

The web-based survey also collected information regarding activity-travel related behaviors, companions, perceptions and attitudes towards different types of elements (such as environmental concerns or use of information and communications technologies), which are out of the scope of this study [63].

## 2.2. Participants

The dataset is composed by 1254 valid responses. Only those survey responses that contained complete information regarding demographic and socioeconomic characteristics, intentions and current use of travel modes, attitudes and values were considered valid. As shown in Table 2, the distribution of the sample according to gender and education level is reasonably balanced. Considering age, those over 70 years old are under-represented in the sample. Taking into account the occupation status, participants are predominantly employed and students.

**Table 2.** Sample distribution.

|  |  | Sample | Valencia Area |
| --- | --- | --- | --- |
| Gender | Male | 48% (596) | 48% |
|  | Female | 52% (658) | 52% |
| Age | <35 | 49% (616) | 36% |
|  | 35–50 | 30% (374) | 25% |
|  | 50–70 | 20% (252) | 25% |
|  | ≥70 | 1% (12) | 14% |
| Occupation | Student | 24% (299) | 5% |
|  | Employed | 55% (690) | 25% |
|  | Others | 21% (257) | 70% |
| Education level | University | 54% (668) | 58% |
|  | No university | 46% (580) | 42% |

## 2.3. Measures

The Schwartz Value Survey (SVS), which is based on the Schwartz theory of human values [16], was used in the web-based survey in order to gather information regarding personal values. In particular, we used the Spanish version of SVS [64], which is composed by 56 items, each one followed by a brief description for clarification. The survey evaluates 10 different value types and four values of higher order types, although values can be clustered in various ways to form a different number of higher order values [65]. Responses are measured on a non-symmetrical scale from −1 to 7 indicating the importance of this value as a guiding principle in the user's life (−1 = opposed to my values, 0 = not important, 3 = important, 6 = very important, and 7 = of supreme importance). The asymmetry of the scale reflects distinctions that people usually make among values [66] and, as the ranking of −1 is quite rare, there is minimal danger to the assumptions of the interval scales [67]. The reliability and validity of the SVS have been demonstrated in several works [16,64,68]. Table 3 shows descriptive statistics as well as the number of items used to measure them. The average rate of universalism and security are higher than the rest, while power and stimulation got the lower scores.

**Table 3.** Personal values.

| Personal Value | Descriptive Statistics | | | | # Items |
| --- | --- | --- | --- | --- | --- |
|  | Mean | Mode | Median | SD |  |
| Universalism | 5.1 | 7 | 5 | 1.6 | 9 |
| Self-direction | 5.2 | 7 | 5 | 1.5 | 7 |
| Stimulation | 3.9 | 4 | 4 | 1.6 | 3 |
| Hedonism | 5.0 | 5 | 5 | 1.5 | 2 |
| Achievement | 4.5 | 5 | 5 | 1.7 | 5 |
| Power | 3.2 | 3 | 3 | 1.9 | 5 |
| Security | 4.8 | 7 | 5 | 1.8 | 7 |
| Tradition | 3.7 | 5 | 4 | 2.3 | 5 |
| Conformity | 4.9 | 5 | 5 | 1.6 | 4 |
| Benevolence | 4.9 | 5 | 5 | 1.7 | 9 |

To gather information regarding the intention to use each travel mode, the following general question was asked to participants: "Which would you like to be your use of each travel mode (in percentage)?" This question collects data regarding desires. But in this study intentions and desires are very similar, because we are studying general mobility. Intentions and desires would be different if the context would have been more specific, for example in terms of time frame [69]. Information regarding the intention to use each travel mode was obtained using a one-hundred-point scale. Thus, participants were asked to distribute 100 points between their intentions to travel by each travel mode (car, carpooling, public transport, walking and cycling). Similarly, the same question was proposed for stating their current use of each travel mode. Thus, both the intention and the current use were obtained and measured with percentages of each mode compared to the total amount of travel. In the cases where the total percentage did not sum to 100 points, a correction was applied in order to standardize the responses and distribute the sum among one hundred percent.

Table 4 shows the mean and median statistics both for intention and current use of each travel mode. As it can be seen, the average intention to use cars (24.6%) is clearly lower than the stated use of cars (33.4%). Conversely, the intention to cycle (18.5%) is notably higher compared to its final use (10.8%). In the case of walking, the difference is smaller but in the same direction. Therefore, intentions to use sustainable transport modes are higher than their current use. This fact might be related to the existence of barriers which could influence behavior.

To evaluate attitudes towards car, public transport, cycling and walking, 16 items were included in the survey for each travel mode. These questions are based in the three-component attitudes model (affective, cognitive and behavioral) and are assessed through 1 to 5-point Likert scales. In general, attitudes towards car and walking are higher than those related to public transport and cycling (Table 5).

**Table 4.** Intention and current use.

| Mode of Transport | Intention to Use (%) | | Stated Use (%) | |
|---|---|---|---|---|
| | **Mean** | **Median** | **Mean** | **Median** |
| Car | 24.6 | 20.0 | 33.4 | 27.3 |
| Carpooling | 5.1 | 0.0 | 5.3 | 0.0 |
| Public transport | 21.0 | 20.0 | 22.6 | 18.1 |
| Bike | 18.5 | 10.0 | 10.8 | 5.0 |
| Walking | 30.8 | 25.0 | 27.9 | 20.0 |

**Table 5.** Attitudes towards travel modes.

| Variable | Descriptive Statistics | | | |
|---|---|---|---|---|
| | **Mean** | **Mode** | **Median** | **SD** |
| Attitudes toward car | 3.4 | 5 | 4 | 1.4 |
| Attitudes toward carpooling | 2.8 | 3 | 3 | 1.3 |
| Attitudes toward public transport | 3.4 | 3 | 3 | 1.2 |
| Attitudes toward cycling | 3.1 | 3 | 3 | 1.4 |
| Attitudes toward walking | 3.8 | 5 | 4 | 1.3 |

## 3. Data Analysis and Results

This section presents data analysis and results of Pearson correlations, Mann–Whitney tests, and hierarchical regression analysis. Pearson correlations are useful to confirm linear relationships between variables. Mann–Whitney tests are used to compare differences between two independent groups when the dependent variable is not normally distributed. Hierarchical regression is a way to show if the variables of interest explain a statistically significant amount of variance in the dependent variable after accounting for all other variables. Other traditional methods of analysis of latent variables like exploratory factor analysis and structural equation models are not recommended to study data

collected through the SVS, because those methods are "not suitable for discovering a set of relations among variables that form a circumflex, as the values data do. The first unrotated factor represents scale use or acquiescence. It is not a substantive common factor." [70].

To avoid a too long paper, the study concentrates only on the sustainable travel modes cycling and walking.

### 3.1. Correlations Between Personal Values, Attitudes, Intentions and Current Use

Pearson's correlation matrices are assessed in order to explore the correlations among the variables of the study for cycling and walking. Considering that attitudes and values are assessed using categorical responses, correlations are obtained using the median value of the set of items for each type of attitudes and values. Median values are used because data regarding attitudes and values do not follow the normal distribution.

Table 6 presents correlations between the ten personal values defined by Schwartz [8] and attitudes, intentions and current use of cycling and walking. The higher number of statistically significant correlations appears between values and attitudes and values and intentions. And the lower number of statistically significant correlations appears between values and current use. These results partially confirm the hierarchy proposed in the conceptual framework.

**Table 6.** Correlations (Pearson) between values and attitudes, intentions and current cycling and walking.

| Values | Attitudes | | Intentions | | Current use | |
|---|---|---|---|---|---|---|
| | **Cycling** | **Walking** | **Cycling** | **Walking** | **Cycling** | **Walking** |
| Universalism | 0.09 *** | 0.17 *** | 0.05 * | −0.01 | 0.03 | 0.03 |
| Self-direction | 0.00 | 0.11 *** | −0.02 | −0.03 | −0.02 | 0.02 |
| Stimulation | 0.21 *** | 0.05 * | 0.12 *** | −0.14 *** | 0.09 *** | −0.09 *** |
| Hedonism | 0.05 * | 0.08 *** | −0.01 | −0.07 ** | 0.00 | −0.06 ** |
| Achievement | 0.04 | 0.02 | −0.04 | −0.10 *** | −0.03 | −0.05 * |
| Power | 0.01 | −0.06 * | −0.09 *** | −0.10 *** | −0.07 ** | −0.10 *** |
| Security | −0.03 | 0.09 *** | −0.14 *** | −0.01 | −0.11 *** | −0.01 |
| Tradition | −0.01 | 0.03 | −0.11 *** | 0.02 | −0.04 | −0.04 |
| Conformity | 0.00 | 0.11 *** | −0.10 *** | 0.00 | −0.07 *** | −0.01 |
| Benevolence | 0.02 | 0.16 *** | −0.08 *** | 0.03 | −0.08 *** | 0.04 |

Note: * $p < 0.1$, ** $p < 0.05$, *** $p < 0.01$.

As expected, attitudes, intentions and current use are significantly and positively correlated for the sustainable modes considered. Intentions and travel behavior present the higher correlations, especially for bicycles.

Correlations between values and the other variables are reasonably low because personal values are general ideas that guide people's life, but attitudes, intentions and current use are referring to particular travel modes. Many statistically significant correlations were obtained, and some important differences can be highlighted. All statistically significant correlations found between values related to self-enhancement and conservation and attitudes, intentions and current cycling and walking are negative. On the other hand, some values related to openness to change and self-transcendence correlated positively and other negatively with attitudes, intentions and current cycling and walking. Stimulation positively correlates with intentions and current use of bicycles, but negatively correlates with intentions and current walking.

### 3.2. Non-Parametric Tests

Mann–Whitney *U* tests for independent samples were used to check significant differences between personal values distribution functions of two or more samples according to demographics and socioeconomics.

Table A1 in Appendix A shows central tendency statistics of the score of each personal value according to respondents' gender. Women present significantly higher scores on universalism, self-direction, hedonism, security and benevolence.

Table A2 in Appendix A shows central tendency statistics of the score of each personal value according to respondents' age. Ages older than 40 score higher on universalism, security, tradition, conformity and benevolence. Ages lower than 40 score higher on stimulation, hedonism and achievement. Table A3 in Appendix A shows central tendency statistics of the score of each personal value according to respondents' income. Those with an income higher than 1000 euro/month present higher significant scores on power, security, tradition and conformity. And those with an income lower than 1000 euro/month (nearly half of them are students) score higher on stimulation, hedonism and achievement. Table A4 in Appendix A shows central tendency statistics of the score of each personal value according to respondents' activity status. Students present significantly higher scores on self-direction, stimulation, hedonism and achievement. On the other hand, workers present higher scores on power and tradition.

### 3.3. Hierarchical Regressions

Hierarchical regressions are fitted in three steps. The first step only includes personal values as explanatory variables, and results are presented in Model 1. The second step includes personal values and attitudes, and results are presented in Model 2. The third step includes personal values, attitudes and intentions, and results are presented in Model 3.

### 3.3.1. Hierarchical Regressions to Predict the Current Cycling and Walking

Hierarchical regressions analyses were performed using current cycling and walking as dependent variables and values, attitudes and intentions as independent variables. First, personal values were entered in the regression model, followed by attitudes and then intentions.

The first step of the model to predict the current cycling was statistically significant ($F_{(10,1240)}$ = 4.46; $p < 0.00$) and explained 4% of the variance (Table A5 in Appendix B). Universalism and stimulation were positive and significant predictors, while power, security and benevolence were negative and significant predictors. The second step, in which the attitude toward cycling was introduced, was also significant ($F_{(1,1239)}$ = 37.74; $p < 0.00$), adding an additional percentage of 22% to the variance of the current cycling. The attitude toward cycling was a positive and significant predictor of the current cycling. In this second step universalism and stimulation lose their significant effect when introducing the attitude toward cycling, which shows that the attitude toward cycling totally mediates the effect of universalism and stimulation on the current cycling. However, power, security and benevolence values maintain their significant effects on the current cycling. The third step, in which the intention to cycle was introduced, was also significant ($F_{(1,1238)}$ = 112.91; $p < 0.00$), adding an additional percentage of 27% to the variance of the current cycling. The intention to cycle was a positive and significant predictor of the current cycling. In this third step power and security values lose their significant effects when introducing the intention to cycle, which shows that the intention to cycle totally mediates the effect of power and security on the current cycling. But tradition gains a significant effect on the current cycling. Benevolence value maintains it significant effect on the current cycling. The attitude toward cycling on the current cycling is statistically significant as well, but to a lower extent, which indicates that intention to cycle partially mediates the effect of the attitudes towards cycling on the current cycling. The model as a whole explained a 52% of the variance of the current use of bicycle.

On the other hand, the first step of the model to predict the current walking was statistically significant ($F_{(10,1241)}$ = 3.15; $p < 0.00$) and explained 2% of the variance (Table A6 in Appendix B). Benevolence was a positive and significant predictor, while stimulation, hedonism and power were negative and significant predictors. The second step, in which the attitude toward walking was introduced, was also significant ($F_{(1,1240)}$ = 14.26; $p < 0.00$), adding an additional percentage of 9%

to the variance of the current walking. The attitude toward walking was a positive and significant predictor of the current walking. In this second step benevolence and power lose their significant effects when introducing the attitude toward walking, which shows that the attitude toward walking totally mediates the effects of benevolence and power on the current walking. However, stimulation and hedonism maintain their negative and significant effect on the current walking. The third step, in which the intention to walk was introduced, was also significant (F (1,1239) = 92.08; $p < 0.00$), adding an additional percentage of 36% to the variance of the current walking. The intention to walk was a positive and significant predictor of the current walking. In this third step stimulation loses its significant effect when introducing the intention to walk, which shows that the intention to walk totally mediates the effect of stimulation on the current walking. However, tradition gains a negative and significant effect and self-direction gains a positive and significant effect on the current walking. Hedonism maintains its significant effect on the current walking. The attitude toward walking on the current walking is statistically significant as well, but to a lower extent, which indicates that intention to walk partially mediates the effect of the attitudes towards walking on the current walking. The model as a whole explained a 47% of the variance of the current walking.

### 3.3.2. Hierarchical Regressions to Predict the Intentions to Cycle and Walk

Hierarchical regressions analyses were performed using intentions to cycle and walk as dependent variables and values and attitudes as independent variables. First, personal values were entered in the regression model, followed by attitudes.

The first step of the model to predict the intention to cycle was statistically significant (F (10,1240) = 7.71; $p < 0.00$) and explained 6% of the variance (Table A7 in Appendix B). Universalism and stimulation were positive and significant predictors, while power and security were negative and significant predictors. The second step, in which the attitudes toward cycling was introduced, was also significant (F (1,1239) = 51.26; $p < 0.00$), adding an additional percentage of 25% to the variance of the intention to cycle. The attitude toward cycling was a positive and significant predictor of the intention to cycle. In this second step stimulation loses its significant effect when introducing the attitude toward cycling, which shows that the attitude toward cycling totally mediates the effect of stimulation on the intention to cycle. However, universalism maintains its positive and significant effect, and power and security maintain their negative and significant effects on the intention to cycle. Benevolence gains a negative and significant effect on the intention to cycle. The model as a whole explained a 31% of the variance of the intention to cycle. The first step of the model to predict the intention to walk was statistically significant (F (10,1241) = 4.25; $p < 0.00$) and explained 3% of the variance (Table A8 in Appendix B). Benevolence was a positive and significant predictor, while stimulation, achievement and power were negative and significant predictors. The second step, in which the attitude towards walking was introduced, was also significant (F (1,1240) = 11.65; $p < 0.00$), adding an additional percentage of 6% to the variance of the intention to walk. The attitude towards walking was a positive and significant predictor of walking. In this second step benevolence and power lose their significant effects when introducing the attitude towards walking, which shows that the attitude towards walking totally mediates the effect of benevolence and power on the intention to walk. However, stimulation and achievement maintain their negative and significant effect on the intention to walk and tradition gains a positive and significant effect on the intention to walk. The model as a whole explained a 9% of the variance of the intention to walk.

### 3.3.3. Regressions to Predict the Attitudes Towards Cycling and Walking

Finally, regression analyses were performed using attitudes towards cycling and walking as dependent variables and personal values as independent variables.

The model to predict the attitude toward cycling was statistically significant (F (10,1240) = 7.36; $p < 0.00$) and explained 6% of the variance (Table A9 in Appendix B). Universalism and stimulation were positive and significant predictors and self-direction was a negative and significant predictor.

The model to predict the attitude toward walking was statistically significant (F (10,1241) = 5.45; $p < 0.00$) and explained 4% of the variance (Table A10 in Appendix B). Universalism and benevolence were positive and significant predictors and achievement was a negative and significant predictor.

## 4. Discussion

The statistically significant estimated correlations between the ten personal values proposed by Schwartz, the attitudes towards cycling and walking and intentions to cycle and walk, are low. Similarly, the correlations between values and current cycling and walking are low as well. Besides, the percentage of explained variance of the predicted use of cycling and walking by the ten personal values is also low. These results are explained because personal values are the basis of all human behaviors [15,16], so they are very general. But both attitudes and intentions need to be related to a specific object. In this case, attitudes and intentions are both related to specific travel modes: Cycling and walking for transport. Additionally, results of the hierarchical regressions indicate that there is a mediation effect between values and current use through attitudes and intentions, which can contribute to the low percentage of explained variance of current cycling and walking by personal values as well.

Many statistically significant estimated correlations and regression coefficients were found. The number of correlations between the ten personal values proposed by Schwartz and attitudes towards cycling and walking is equal to the number of correlations between the ten personal values and intentions to cycle and walk. And they both are higher than the number of correlations between the ten personal values and the current cycling and walking. This result is an indication that the association between values and attitudes and intentions is stronger than the influence of values on behavior, which confirms our conceptual framework.

Results from the hierarchical regressions denote that the explained variance of intentions to cycle by personal values is higher than the explained variance of current cycling. And the explained variance of attitudes towards cycling and walking by personal values is higher than the explained variance of intentions to cycle and walk. Therefore, besides indirect effects, there are significant direct effects between values and the other variables considered.

Additionally, results from the hierarchical regressions indicate that the attitudes toward cycling totally mediates the effect of universalism and stimulation on current use of bicycle. And the intention to cycle totally mediates the effect of power and security on current use of bicycle. The attitudes toward cycling totally mediates the effect of stimulation on the intention to cycle. Similarly, attitudes toward walking totally mediates the effect of benevolence and power on both the intention to walk and current walking. And the intention to walk totally mediates the effect of stimulation on the current walking.

The results described above suggest that the conceptual framework proposed in this study that links psychological variables with travel behavior needs to be improved as indicated in Figure 2.

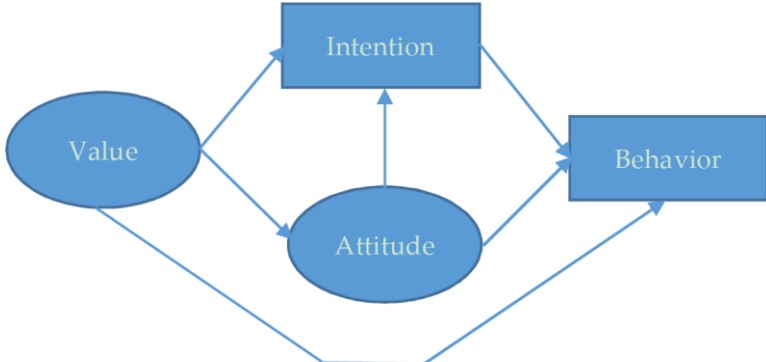

**Figure 2.** Improved conceptual framework.

Our hypotheses regarding the sign of the effect of personal values on attitudes, intentions and current cycling and walking are partially confirmed. Regarding openness to change related values

(self-direction and stimulation) and self-transcendence related values (benevolence and universalism), universalism and stimulation personal values positively influence cycling on the three variables (attitude, intention and current use), but we have found a negative influence of self-direction on the attitude toward cycling and a negative influence of benevolence on the current cycling. Cycling is not associated to any direct benefit to others, but their own benefit is especially related to experimentation and an exciting and varied life. It is also possible that those benefits were also connected to convenience because the city of Valencia is increasing both the extension and quality of its bike lanes network. Helpful and responsible people are more sensible to the needs of others, and more likely to not use bicycles to escort siblings, colleagues or friends.

Regarding openness to change related values (self-direction and stimulation) and self-transcendence related values (benevolence and universalism), benevolence positively influence walking on the three variables (attitude, intention and current use), universalism positively influence the attitude toward walking and we have found a negative influence of stimulation on the intention to walk and the current walking. On the other hand, walking (for transport) is not contemplated as exciting or daring. The modal share of walking in the city of Valencia, as in many others Spanish cities, is quite high (around 40% [71]), which means that people is get used to walk to carry out any type of activity.

Regarding self-enhancement related values (hedonism, achievement and power) and conservation (security, conformity and tradition) as it was hypothesized that power and security personal values have a negative influence on intentions to cycle and current cycling, in line with the findings of Hunecke et al. [22]. On the other hand, hedonism and power negatively influence the current walking and power also negatively influences the intention to walk. In addition, achievement negatively influences the attitude towards walk and the intention to walk. It seems that people do not associate the prestige and social status, the stability of society, relationships and stability of the self, a sense of gratification for oneself or a feeling of personal success when cycling or walking.

Finally, socioeconomic and demographic variables also matter. Women present significantly higher scores on universalism, self-direction, hedonism, security and benevolence. Except for security, these values are positively associated to walking and cycling.

Those older than 40 score higher on universalism, security, tradition, conformity and benevolence. Those younger than 40 score higher on stimulation, hedonism and achievement. In both cases, there is a mix of values positively and negatively associated to walking and cycling.

Those with an income higher than 1000 euro/month present higher significant scores on power, security, tradition and conformity. Those with an income lower than 1000 euro/month (nearly half of them are students) score higher on stimulation, hedonism, and achievement. The former are negatively aligned with walking and cycling, and the latter are positively aligned with walking and cycling.

Students present significantly higher scores on self-direction, stimulation, hedonism and achievement, which are values positively associated with cycling and walking. On the other hand, workers present higher scores on power and tradition, which are values negatively associated to cycling and walking.

## 5. Conclusions

This study presents the findings of an analysis of the influence of personal values on attitudes towards cycling and walking, intentions and real use of cycling and walking. As expected, openness to change related values are significantly and positively associated with cycling and walking. And self-enhancement related values are significantly and negatively associated with cycling and walking.

Although personal values are relatively stable over time, it is possible to encourage their change. For example, techniques based on disseminating successful experiences have been proved to be effective [72].

Value transmission from one generation to the next occurs to a greater extent at family level. In this context, encouraging self-transcendence values (helping, supporting and caring for others), which are positively associated to cycling and walking, is effective if parents also share the same personal

values [73]. Therefore, it is important that Travel Behavior Change Programs (TBCP) include actions and activities to encourage self-transcendence values in which all family members are involved.

The promotion of positive attitudes towards cycling and walking is important because they mediate between openness to change (motivating people to be independent, curious and experience a varied life) and self-transcendence values, and current cycling and walking. In this regard, there are many strategies to achieve that goal. For example, implementing policies that aim at removing barriers perceived as external [45] or awareness campaigns to inform of health benefits of cycling and walking [74].

Females and those over 40 years old are more likely to score higher on self-transcendence values. Although those younger than 40 also score higher on openness to change and self-enhancement values, as well as students and those with an income lower than 1000 euro/month. Actions to promote cycling and walking could be more effective including activities to encourage self-transcendence or openness to change values considering those demographic and socioeconomic profiles.

On the other hand, the survey method used in this research caused that those over 70 years old are under-represented in the sample, and participants were predominantly employed and students. We acknowledge that our study only represents those younger than 70 years old, students and employed people, which is a limitation of this research

Finally, this study can be extended in a number of ways. Only nonmotorized travel modes have been considered in the analyses presented in this paper. We plan to investigate how personal values affect attitudes towards car and public transport, intentions to drive and use of public transport, and current use of car and public transport. It would also be interesting to assess the potential mediation effects of attitudes and intentions associated with a particular travel mode between values and another travel mode. To this end, other modelling tools will be necessary.

To confirm the findings included in this paper, personal values could be investigated in the context of travel behavior, so that stronger relationships could be detected. For this purpose, an adaptation of the Schwartz scale would be needed.

**Author Contributions:** Conceptualization, J.G., L.M., F.d.C. and T.R.; formal analysis, J.G., L.M., R.A. and T.R.; funding acquisition, T.R.; investigation, J.G. and L.M.; methodology, L.M., R.A. and D.C.; software, R.A. and D.C.; supervision, L.M. and T.R.; validation, R.A. and F.d.C.; writing—original draft, J.G., L.M. and R.A.; and writing—review and editing, J.G., L.M., F.d.C. and T.R.

**Funding:** This study has been partially funded by the Ministerio de Economía, Industria y Competitividad of Spain, project MINERVA (TRA2015-71184-C2-1-R).

**Conflicts of Interest:** The authors declare no conflict of interest.

## Appendix A. Non-Parametric Tests for Personal Values

**Table A1.** Nonparametric test for personal values and gender.

|  | Gender | N | Mean | Median | Mode | SD | Mann–Whitney Test | | |
|---|---|---|---|---|---|---|---|---|---|
|  |  |  |  |  |  |  | Mean Rank | Z | Sign. |
| Universalism | Male | 591 | 5.09 | 4.00 | 5.00 | 1.2 | 584.9 | −3.82 | 0.00 *** |
|  | Female | 658 | 5.36 | 4.00 | 5.00 | 1.2 | 661.0 |  |  |
| Self-direction | Male | 591 | 5.27 | 5.00 | 5.00 | 1.2 | 596.3 | −2.74 | 0.01 *** |
|  | Female | 658 | 5.47 | 5.00 | 5.00 | 1.1 | 650.8 |  |  |
| Stimulation | Male | 591 | 3.93 | 6.00 | 7.00 | 1.4 | 616.2 | −0.79 | 0.43 |
|  | Female | 658 | 4.01 | 6.00 | 7.00 | 1.5 | 631.9 |  |  |
| Hedonism | Male | 591 | 4.90 | 5.00 | 5.00 | 1.2 | 604.0 | −1.97 | 0.05 ** |
|  | Female | 658 | 5.04 | 5.00 | 5.00 | 1.2 | 643.9 |  |  |
| Achievement | Male | 591 | 4.46 | 5.00 | 5.00 | 1.4 | 615.3 | −0.93 | 0.36 |
|  | Female | 658 | 4.55 | 5.00 | 5.00 | 1.3 | 633.8 |  |  |

**Table A1.** *Cont.*

| | Gender | N | Mean | Median | Mode | SD | Mann–Whitney Test | | |
| --- | --- | --- | --- | --- | --- | --- | --- | --- | --- |
| | | | | | | | Mean Rank | Z | Sign. |
| Power | Male | 591 | 3.19 | 5.00 | 5.00 | 1.5 | 622.9 | −0.21 | 0.84 |
| | Female | 658 | 3.20 | 5.00 | 5.00 | 1.4 | 626.9 | | |
| Security | Male | 591 | 4.87 | 4.00 | 4.00 | 1.2 | 598.8 | −2.50 | 0.01 *** |
| | Female | 658 | 5.06 | 4.00 | 5.00 | 1.2 | 648.5 | | |
| Tradition | Male | 591 | 4.02 | 4.00 | 4.00 | 1.5 | 634.3 | −0.88 | 0.38 |
| | Female | 658 | 3.96 | 4.00 | 4.00 | 1.5 | 616.7 | | |
| Conformity | Male | 591 | 4.96 | 3.50 | 3.00 | 1.2 | 615.9 | −0.85 | 0.40 |
| | Female | 658 | 5.01 | 3.50 | 3.00 | 1.2 | 633.2 | | |
| Benevolence | Male | 591 | 4.95 | 5.00 | 5.00 | 1.2 | 581.1 | −4.21 | 0.00 *** |
| | Female | 658 | 5.24 | 5.00 | 4.00 | 1.1 | 664.5 | | |

Note: * $p < 0.1$, ** $p < 0.05$, *** $p < 0.01$.

**Table A2.** Nonparametric tests for personal values and age.

| Values | Age | N | Mean | Median | Mode | SD | Mann–Whitney Test | | |
| --- | --- | --- | --- | --- | --- | --- | --- | --- | --- |
| | | | | | | | Mean Rank | Z | Sign. |
| Universalism | <40 | 723 | 5.14 | 5.0 | 5.0 | 1.22 | 599.1 | −3.32 | 0.00 *** |
| | ≥40 | 531 | 5.37 | 5.5 | 6.0 | 1.26 | 666.1 | | |
| Self-direction | <40 | 723 | 5.36 | 5.0 | 5.0 | 1.13 | 623.2 | -0.51 | 0.61 |
| | ≥40 | 531 | 5.40 | 5.0 | 5.0 | 1.20 | 633.4 | | |
| Stimulation | <40 | 723 | 4.16 | 4.0 | 5.0 | 1.41 | 672.8 | −5.35 | 0.00 *** |
| | ≥40 | 530 | 3.72 | 4.0 | 3.0 | 1.47 | 564.6 | | |
| Hedonism | <40 | 723 | 5.06 | 5.0 | 5.0 | 1.16 | 654.3 | −3.08 | 0.00 *** |
| | ≥40 | 531 | 4.87 | 5.0 | 5.0 | 1.20 | 591.0 | | |
| Achievement | <40 | 723 | 4.61 | 5.0 | 5.0 | 1.35 | 657.0 | −3.45 | 0.00 *** |
| | ≥40 | 531 | 4.37 | 4.0 | 5.0 | 1.30 | 587.3 | | |
| Power | <40 | 723 | 3.19 | 3.0 | 3.0 | 1.50 | 624.9 | −0.30 | 0.76 |
| | ≥40 | 531 | 3.21 | 3.0 | 3.0 | 1.48 | 631.0 | | |
| Security | <40 | 723 | 4.89 | 5.0 | 5.0 | 1.17 | 604.5 | −2.70 | 0.01 *** |
| | ≥40 | 531 | 5.08 | 5.0 | 5.0 | 1.30 | 658.8 | | |
| Tradition | <40 | 723 | 3.79 | 4.0 | 3.0 | 1.54 | 581.7 | −5.34 | 0.00 *** |
| | ≥40 | 531 | 4.26 | 4.0 | 3.0 | 1.40 | 689.8 | | |
| Conformity | <40 | 723 | 4.85 | 5.0 | 5.0 | 1.26 | 589.5 | −4.37 | 0.00 *** |
| | ≥40 | 531 | 5.17 | 5.0 | 5.0 | 1.20 | 679.3 | | |
| Benevolence | <40 | 723 | 5.02 | 5.0 | 5.0 | 1.13 | 601.9 | −3.02 | 0.00 *** |
| | ≥40 | 531 | 5.22 | 5.0 | 5.0 | 1.19 | 662.4 | | |

Note: * $p < 0.1$, ** $p < 0.05$, *** $p < 0.01$.

**Table A3.** Nonparametric test for personal values and income.

| Values | Income (€) | N | Mean | Median | Mode | SD | Mann–Whitney Test | | |
| --- | --- | --- | --- | --- | --- | --- | --- | --- | --- |
| | | | | | | | Mean Rank | Z | Sign. |
| Universalism | ≤1000 | 573 | 4.38 | 5.00 | 5.00 | 0.8 | 629.5 | −1.0 | 0.33 |
| | >1000 | 664 | 4.26 | 4.00 | 5.00 | 0.9 | 610.0 | | |
| Self-direction | ≤1000 | 573 | 4.44 | 5.00 | 5.00 | 2.3 | 635.0 | −1.5 | 0.13 |
| | >1000 | 664 | 4.24 | 4.00 | 5.00 | 1.8 | 605.2 | | |

**Table A3.** *Cont.*

| Values | Income (€) | N | Mean | Median | Mode | SD | Mann–Whitney Test | | |
|---|---|---|---|---|---|---|---|---|---|
| | | | | | | | Mean Rank | Z | Sign. |
| Stimulation | ≤1000 | 573 | 6.36 | 6.00 | 7.00 | 3.6 | 645.8 | −2.6 | 0.01 *** |
| | >1000 | 663 | 6.08 | 6.00 | 7.00 | 1.9 | 594.9 | | |
| Hedonism | ≤1000 | 573 | 5.53 | 5.50 | 7.00 | 1.2 | 639.0 | −1.8 | 0.07 * |
| | >1000 | 664 | 5.54 | 6.00 | 7.00 | 1.3 | 601.7 | | |
| Achievement | ≤1000 | 573 | 5.12 | 5.00 | 5.00 | 1.4 | 652.9 | −3.2 | 0.00 *** |
| | >1000 | 664 | 5.10 | 5.00 | 6.00 | 1.4 | 589.7 | | |
| Power | ≤1000 | 573 | 5.55 | 6.00 | 6.00 | 1.2 | 591.6 | −2.6 | 0.01 *** |
| | >1000 | 664 | 5.42 | 6.00 | 6.00 | 1.2 | 642.6 | | |
| Security | ≤1000 | 573 | 5.16 | 5.00 | 5.00 | 1.1 | 593.7 | −2.4 | 0.02 ** |
| | >1000 | 664 | 5.06 | 5.00 | 5.00 | 1.2 | 640.9 | | |
| Tradition | ≤1000 | 573 | 4.42 | 4.50 | 5.00 | 1.3 | 571.6 | −4.4 | 0.00 *** |
| | >1000 | 664 | 4.22 | 4.00 | 4.00 | 1.3 | 659.9 | | |
| Conformity | ≤1000 | 573 | 4.79 | 5.00 | 5.00 | 1.2 | 600.5 | −1.7 | 0.09 * |
| | >1000 | 664 | 4.62 | 4.50 | 5.00 | 1.2 | 634.9 | | |
| Benevolence | ≤1000 | 573 | 3.85 | 4.00 | 4.00 | 1.3 | 617.5 | −0.1 | 0.88 |
| | >1000 | 664 | 3.89 | 4.00 | 3.00 | 1.3 | 620.3 | | |

Note: * $p < 0.1$, ** $p < 0.05$, *** $p < 0.01$.

**Table A4.** Nonparametric test for personal values and occupational status.

| Values | Activity Status | N | Mean | Median | Mode | SD | Mann–Whitney Test | | |
|---|---|---|---|---|---|---|---|---|---|
| | | | | | | | Mean Rank | Z | Sign. |
| Universalism | Student | 299 | 5.20 | 5.00 | 5.00 | 1.22 | 502.3 | −0.39 | 0.70 |
| | Worker | 694 | 5.16 | 4.00 | 5.00 | 1.23 | 494.7 | | |
| Self-direction | Student | 299 | 5.47 | 5.00 | 5.00 | 1.07 | 525.3 | −2.11 | 0.04 ** |
| | Worker | 694 | 5.30 | 4.00 | 5.00 | 1.19 | 484.8 | | |
| Stimulation | Student | 299 | 4.17 | 6.00 | 7.00 | 1.45 | 524.5 | −2.07 | 0.04 ** |
| | Worker | 694 | 3.98 | 6.00 | 7.00 | 1.38 | 484.4 | | |
| Hedonism | Student | 299 | 5.05 | 6.00 | 7.00 | 1.13 | 520.2 | −1.69 | 0.09 * |
| | Worker | 694 | 4.92 | 5.50 | 7.00 | 1.19 | 487.0 | | |
| Achievement | Student | 299 | 4.74 | 5.00 | 5.00 | 1.34 | 543.0 | −3.39 | 0.00 *** |
| | Worker | 694 | 4.42 | 5.00 | 5.00 | 1.34 | 477.2 | | |
| Power | Student | 299 | 3.08 | 5.00 | 5.00 | 1.47 | 466.8 | −2.24 | 0.03 ** |
| | Worker | 694 | 3.30 | 5.00 | 5.00 | 1.46 | 510.0 | | |
| Security | Student | 299 | 4.86 | 5.50 | 6.00 | 1.11 | 478.5 | −1.37 | 0.17 |
| | Worker | 694 | 4.96 | 5.00 | 6.00 | 1.25 | 505.0 | | |
| Tradition | Student | 299 | 3.63 | 4.00 | 5.00 | 1.48 | 428.5 | −5.05 | 0.00 *** |
| | Worker | 694 | 4.15 | 4.00 | 4.00 | 1.48 | 526.5 | | |
| Conformity | Student | 299 | 4.84 | 4.50 | 4.00 | 1.31 | 478.6 | −1.34 | 0.18 |
| | Worker | 694 | 4.99 | 4.50 | 4.00 | 1.21 | 504.9 | | |
| Benevolence | Student | 299 | 5.04 | 4.00 | 3.00 | 1.14 | 495.5 | −0.11 | 0.91 |
| | Worker | 694 | 5.07 | 4.00 | 3.00 | 1.17 | 497.6 | | |

Note: * $p < 0.1$, ** $p < 0.05$, *** $p < 0.01$.

## Appendix B. Regression Models

**Table A5.** Hierarchical regression to predict the current cycling.

|  | $R^2$ | $\Delta R^2$ | B | SE | $\beta$ | $t$ | **Sig.** |
|---|---|---|---|---|---|---|---|
| **Model 1** | **0.04** | **0.04** |  |  |  |  |  |
| Universalism |  |  | 1.10 | 0.53 | 0.08 | 2.09 ** | 0.04 |
| Self-direction |  |  | −0.20 | 0.61 | −0.01 | −0.32 | 0.75 |
| Stimulation |  |  | 1.36 | 0.39 | 0.12 | 3.52 *** | 0.00 |
| Hedonism |  |  | 0.30 | 0.49 | 0.02 | 0.61 | 0.55 |
| Achivement |  |  | 0.09 | 0.49 | 0.01 | 0.18 | 0.86 |
| Power |  |  | −0.79 | 0.39 | −0.07 | −2.02 ** | 0.04 |
| Security |  |  | −1.27 | 0.55 | −0.09 | −2.34 ** | 0.02 |
| Tradition |  |  | 0.27 | 0.39 | 0.02 | 0.69 | 0.49 |
| Conformity |  |  | −0.14 | 0-57 | −0.01 | −0.24 | 0.81 |
| Benevolence |  |  | −1.39 | 0.65 | −0.09 | −2.14 ** | 0.03 |
| **Model 2** | **0.25** | **0.22** |  |  |  |  |  |
| Universalism |  |  | 0.47 | 0.47 | 0.03 | 1.01 | 0.31 |
| Self-direction |  |  | 0.32 | 0.53 | 0.02 | 0.60 | 0.55 |
| Stimulation |  |  | 0.05 | 0.35 | 0.01 | 0.16 | 0.88 |
| Hedonism |  |  | 0.38 | 043 | 0.03 | 0.87 | 0.38 |
| Achivement |  |  | 0.20 | 0.44 | 0.02 | 0.46 | 0.65 |
| Power |  |  | −0.69 | 0.35 | −0.06 | −1.99 ** | 0.05 |
| Security |  |  | −0.98 | 0.48 | −0.07 | −2.04 ** | 0.04 |
| Tradition |  |  | 0.37 | 0.35 | 0.03 | 1.08 | 0.28 |
| Conformity |  |  | −0.22 | 0.51 | −0.02 | −0.44 | 0.66 |
| Benevolence |  |  | −1.34 | 0.57 | −0.09 | −2.34 ** | 0.02 |
| Attitude Bicycle |  |  | 7.31 | 0.39 | 0.48 | 18.91 *** | 0.00 |
| **Model 3** | **0.52** | **0.27** |  |  |  |  |  |
| Universalism |  |  | −0.11 | 0.37 | −0.01 | −0.30 | 0.77 |
| Self-direction |  |  | 0.09 | 0.43 | 0.01 | 0.20 | 0.84 |
| Stimulation |  |  | −0.18 | 0.28 | −0.02 | −0.65 | 0.52 |
| Hedonism |  |  | 0.43 | 0.35 | 0.03 | 1.24 | 0.21 |
| Achivement |  |  | 0.02 | 0.35 | 0.00 | 0.06 | 0.95 |
| Power |  |  | −0.25 | 0.28 | −0.02 | −0.89 | 0.38 |
| Security |  |  | −0.23 | 0.39 | −0.02 | −0.59 | 0.56 |
| Tradition |  |  | 0.68 | 0.28 | 0.06 | 2.46 *** | 0.01 |
| Conformity |  |  | −0.02 | 0.41 | −0.00 | −0.04 | 0.97 |
| Benevolence |  |  | −0.75 | 0.46 | −0.01 | −1.63 * | 0.10 |
| Attitude Bicycle |  |  | 2.33 | 0.36 | 0.15 | 6.44 *** | 0.00 |
| Intention Bicycle |  |  | 0.53 | 0.00 | 0.63 | 26.54 *** | 0.00 |

Note: $R^2$ and $\Delta R^2$ are significant at level 0.01 in all models. * $p < 0.10$, ** $p < 0.05$, *** $p < 0.01$.

**Table A6.** Hierarchical regression to predict the current walking.

|  | $R^2$ | $\Delta R^2$ | B | SE | $\beta$ | t | **Sign.** |
|---|---|---|---|---|---|---|---|
| **Model 1** | **0.03** | **0.03** |  |  |  |  |  |
| Universalism |  |  | 0.22 | 0.65 | 0.01 | 0.34 | 0.73 |
| Self-direction |  |  | 0.92 | 0.74 | 0.05 | 1.23 | 0.22 |
| Stimulation |  |  | −1.06 | 0.48 | −0.07 | −2.24 ** | 0.03 |
| Hedonism |  |  | −1.16 | 0.61 | −0.07 | −1.91 * | 0.06 |
| Achievement |  |  | −0.28 | 0.61 | −0.02 | −0.46 | 0.65 |
| Power |  |  | −0.86 | 0.41 | −0.06 | −1.78 * | 0.08 |
| Security |  |  | 0.24 | 0.60 | 0.01 | 0.35 | 0.73 |
| Tradition |  |  | −0.52 | 0.48 | −0.04 | −1.08 | 0.28 |
| Conformity |  |  | −0.59 | 0.71 | −0.04 | −0.83 | 0.40 |
| Benevolence |  |  | 1.58 | 0.80 | 0.09 | 1.98 ** | 0.05 |

**Table A6.** *Cont.*

|  | $R^2$ | $\Delta R^2$ | B | SE | β | t | Sign. |
|---|---|---|---|---|---|---|---|
| **Model 2** | **0.11** | **0.09** |  |  |  |  |  |
| Universalism |  |  | −0.31 | 0.62 | −0.02 | −0.50 | 0.62 |
| Self-direction |  |  | 0.99 | 0.71 | 0.06 | 1.40 | 0.16 |
| Stimulation |  |  | −1.18 | 0.45 | −0.08 | −2.60 *** | 0.01 |
| Hedonism |  |  | −1.21 | 0.59 | −0.07 | −2.10 ** | 0.04 |
| Achievement |  |  | 0.04 | 0.58 | 0.00 | 0.06 | 0.95 |
| Power |  |  | −0.63 | 0.46 | −0.05 | −1.36 | 0.17 |
| Security |  |  | 0.13 | 0.64 | 0.01 | 0.20 | 0.84 |
| Tradition |  |  | −0.33 | 0.46 | −0.02 | −0.72 | 0.47 |
| Conformity |  |  | −0.88 | 0.68 | −0.05 | −10.30 | 0.20 |
| Benevolence |  |  | 1.03 | 0.76 | 0.06 | 1.35 | 0.18 |
| Attitude Walk |  |  | 7.07 | 0.64 | 0.30 | 11.06 *** | 0.00 |
| **Model 3** | **0.47** | **0.36** |  |  |  |  |  |
| Universalism |  |  | −0.06 | 0.48 | −0.00 | −0.13 | 0.90 |
| Self-direction |  |  | 0.99 | 0.55 | 0.06 | 1.81 * | 0.07 |
| Stimulation |  |  | −0.17 | 0.35 | −0.01 | −0.47 | 0.64 |
| Hedonism |  |  | −0.98 | 0.45 | −0.06 | −2.21 ** | 0.02 |
| Achievement |  |  | 0.64 | 0.45 | 0.04 | 1.44 | 0.15 |
| Power |  |  | −0.22 | 0.36 | −0.02 | −0.62 | 0.54 |
| Security |  |  | 0.11 | 0.49 | 0.01 | 0.22 | 0.82 |
| Tradition |  |  | −0.82 | 0.36 | −0.06 | −2.30 ** | 0.02 |
| Conformity |  |  | −0.58 | 0.52 | −0.04 | −1.12 | 0.27 |
| Benevolence |  |  | 0.30 | 0.59 | 0.02 | 0.56 | 0.58 |
| Attitude Walk |  |  | 3.37 | 0.51 | 0.14 | 6.61 *** | 0.00 |
| Intention Walk |  |  | 0.57 | 0.02 | 0.63 | 29.02 *** | 0.00 |

Note: $R^2$ and $\Delta R^2$ are significant at level 0.01 in all models. * $p < 0.10$, ** $p < 0.05$, *** $p < 0.01$.

**Table A7.** Hierarchical regression to predict the intention to cycle.

|  | $R^2$ | $\Delta R^2$ | B | SE | β | t | Sign. |
|---|---|---|---|---|---|---|---|
| **Model 1** | **0.05** | **0.05** |  |  |  |  |  |
| Universalism |  |  | 1.91 | 0.62 | 0.12 | 3.08 *** | 0.00 |
| Self-direction |  |  | −0.22 | 0.71 | −0.01 | −0.31 | 0.76 |
| Stimulation |  |  | 2.13 | 0.45 | 0.15 | 4.69 *** | 0.00 |
| Hedonism |  |  | −0.20 | 0.58 | −0.01 | −0.35 | 0.73 |
| Achievement |  |  | 0.19 | 0.58 | 0.01 | 0.33 | 0.74 |
| Power |  |  | −0.97 | 0.46 | −0.07 | −2.11 ** | 0.04 |
| Security |  |  | −1.81 | 0.64 | −0.11 | −20.82 *** | 0.01 |
| Tradition |  |  | −0.71 | 0.46 | −0.05 | −1.51 | 0.12 |
| Conformity |  |  | −0.28 | 0.68 | −0.02 | −0.42 | 0.68 |
| Benevolence |  |  | −1.18 | 0.76 | −0.07 | −1.55 | 0.12 |
| **Model 2** | **0.31** | **0.25** |  |  |  |  |  |
| Universalism |  |  | 1.10 | 0.53 | 0.07 | 2.07 ** | 0.04 |
| Self-direction |  |  | 0.44 | 0.61 | 0.03 | 0.73 | 0.47 |
| Stimulation |  |  | 0.45 | 0.40 | 0.03 | 1.12 | 0.26 |
| Hedonism |  |  | −0.10 | 0.50 | −0.01 | −0.20 | 0.84 |
| Achievement |  |  | 0.34 | 0.50 | 0.02 | 0.68 | 0.50 |
| Power |  |  | −0.84 | 0.39 | −0.06 | −2.13 ** | 0.03 |
| Security |  |  | −10.42 | 0.55 | −0.09 | −2.60 *** | 0.01 |
| Tradition |  |  | −0.58 | 0.39 | −0.04 | −1.47 | 0.14 |
| Conformity |  |  | −0.39 | 0.58 | −0.02 | −0.68 | 0.50 |
| Benevolence |  |  | −1.12 | 0.65 | −0.06 | −1.72 * | 0.09 |
| Attitude Bicycle |  |  | 9.41 | 0.44 | 0.52 | 21.41 *** | 0.00 |

Note: $R^2$ and $\Delta R^2$ are significant at level 0.01 in all models. * $p < 0.10$, ** $p < 0.05$, *** $p < 0.01$.

**Table A8.** Hierarchical regression to predict the intention to walk.

|  | $R^2$ | $\Delta R^2$ | B | SE | $\beta$ | $t$ | Sign. |
|---|---|---|---|---|---|---|---|
| **Model 1** | 0.03 | 0.03 |  |  |  |  |  |
| Universalism |  |  | 0.05 | 0.71 | 0.00 | 0.07 | 0.94 |
| Self-direction |  |  | −0.07 | 0.82 | −0.00 | −0.08 | 0.94 |
| Stimulation |  |  | −1.67 | 0.52 | −0.11 | −3.20 ** | 0.00 |
| Hedonism |  |  | −0.34 | 0.66 | −0.02 | −0.52 | 0.60 |
| Achievement |  |  | −1.35 | 0.66 | −0.08 | −2.03 ** | 0.04 |
| Power |  |  | −0.92 | 0.53 | −0.06 | −1.75 * | 0.08 |
| Security |  |  | 0.13 | 0.74 | 0.01 | 0.17 | 0.86 |
| Tradition |  |  | 0.67 | 0.53 | 0.04 | 1.28 | 0.20 |
| Conformity |  |  | −0.25 | 0.78 | −0.01 | −0.33 | 0.74 |
| Benevolence |  |  | 1.73 | 0.87 | 0.09 | 1.98 ** | 0.05 |
| **Model 2** | 0.09 | 0.06 |  |  |  |  |  |
| Universalism |  |  | −0.44 | 0.69 | −0.02 | −0.63 | 0.53 |
| Self-direction |  |  | 0.00 | 0.9 | 0.00 | 0.00 | 0.99 |
| Stimulation |  |  | −1.73 | 0.51 | −0.11 | −3.51 *** | 0.00 |
| Hedonism |  |  | −0.40 | 0.64 | −0.02 | −0.62 | 0.54 |
| Achievement |  |  | −1.06 | 0.64 | −0.06 | −1.65 * | 0.10 |
| Power |  |  | −0.71 | 0.51 | −0.05 | −1.39 | 0.16 |
| Security |  |  | 0.03 | 0.71 | 0.00 | 0.041 | 0.97 |
| Tradition |  |  | 0.85 | 0.51 | 0.06 | 1.66 * | 0.10 |
| Conformity |  |  | −0.52 | 0.75 | −0.03 | −0.69 | 0.49 |
| Benevolence |  |  | 1.22 | 0.85 | 0.06 | 1.44 | 0.15 |
| Attitude Walking |  |  | 6.48 | 0.71 | 0.25 | 9.10 *** | 0.00 |

Note: $R^2$ and $\Delta R^2$ are significant at level 0.01 in all models. * $p < 0.10$, ** $p < 0.05$, *** $p < 0.01$.

**Table A9.** Regression to predict the attitude towards cycling.

|  | $R^2$ | $\Delta R^2$ | B | SE | $\beta$ | $t$ | Sign. |
|---|---|---|---|---|---|---|---|
| **Model 1** | 0.06 | 0.06 |  |  |  |  |  |
| Universalism |  |  | 0.09 | 0.03 | 0.10 | 2.52 *** | 0.01 |
| Self-direction |  |  | −0.07 | 0.04 | −0.07 | −1.80 * | 0.07 |
| Stimulation |  |  | 0.18 | 0.03 | 0.23 | 7.14 *** | 0.00 |
| Hedonism |  |  | −0.01 | 0.03 | −0.01 | −0.35 | 0.73 |
| Achievement |  |  | −0.02 | 0.03 | −0.02 | −0.48 | 0.63 |
| Power |  |  | −0.01 | 0.03 | −0.02 | −0.56 | 0.58 |
| Security |  |  | −0.04 | 0.04 | −0.05 | −1.15 | 0.25 |
| Tradition |  |  | −0.01 | 0.03 | −0.02 | −0.55 | 0.58 |
| Conformity |  |  | 0.01 | 0.04 | 0.01 | 0.31 | 0.75 |
| Benevolence |  |  | −0.01 | 0.04 | −0.01 | −0.15 | 0.88 |

Note: $R^2$ and $\Delta R^2$ are significant at level 0.01. * $p < 0.10$, ** $p < 0.05$, *** $p < 0.01$.

**Table A10.** Regression to predict the attitude towards walking.

|  | $R^2$ | $\Delta R^2$ | B | SE | $\beta$ | $t$ | Sign. |
|---|---|---|---|---|---|---|---|
| **Model 1** | 0.04 | 0.04 |  |  |  |  |  |
| Universalism |  |  | 0.08 | 0.03 | 0.11 | 2.75 *** | 0.01 |
| Self-direction |  |  | −0.01 | 0.03 | −0.01 | −0.34 | 0.74 |
| Stimulation |  |  | 0.02 | 0.02 | 0.03 | 0.80 | 0.42 |
| Hedonism |  |  | 0.01 | 0.03 | 0.01 | 0.31 | 0.75 |
| Achievement |  |  | −0.04 | 0.03 | −0.07 | −1.73 * | 0.09 |
| Power |  |  | −0.03 | 0.02 | −0.05 | −1.59 | 0.11 |
| Security |  |  | 0.02 | 0.03 | 0.02 | 0.54 | 0.59 |
| Tradition |  |  | −0.03 | 0.02 | −0.05 | −1.33 | 0.19 |
| Conformity |  |  | 0.04 | 0.03 | 0.06 | 1.35 | 0.18 |
| Benevolence |  |  | 0.08 | 0.03 | 0.10 | 2.32 ** | 0.02 |

Note: $R^2$ and $\Delta R^2$ are significant at level 0.01. * $p < 0.10$, ** $p < 0.05$, *** $p < 0.01$.

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
