# Peer review of "Personal Values, Attitudes and Travel Intentions Towards Cycling and Walking, and Actual Behavior"

_sustainability, doi:10.3390/su11133574_

Round 1

Reviewer 1 Report

I like the richness of values included, the straightforward methodology making the paper easy to understand, and the conceptual model as presented in Figure 3. But I have some concerns.

General comments

·         There is more literature on values. The paper should make the gap(s) in the literature and the added value of this paper more explicit in the introduction.

·         Why such a large differences (two to nine) in the number of items per value (Table 2)?

·         Table 3: how was intention to use asked for? And: is this about general intentions to use, or for specific cases?

·         It seems that all modes are included in the survey. Why then the focus on cycling and walking, and not an all modes?

·         I like figure 3, for me this is an important contribution of the paper. But if this is how it works, then why not a SEM model?

·         Final section: conclusions and discussion could be distinguished more clearly.

A matter of taste, but it could be an option to move most tables to appendices, and condense the results in the main text.

Details

·         35: ‘However, the sustainable paradigm, the recent economic recession and the new urban mobility solutions are testing this aforementioned hypothesis.’  Why?

·         38: ‘it is well known that ….’: add references.

·         Figure 1 and related text: OK, but there are many more publications that list and cluster values. Why did you choose this one (only)?

·         97: add dot at end sentence.

·         157: May-October: why such a long period for the data collection? And sure, several people will be on holiday in August, but why then close the questionnaire? I would simply expect lower numbers of respondents, but why would there be bias?

·         Table 1: can you compare the sample with the population?

·         179: why a scale ranging from MINUS 1 to seven? Why not 1-7? Which options: integers only? Why non-symmetrical? What is non-symmetrical anyway?

·         Table 2 hedonism: use dot, not comma for the SD.

·         179 and 200: why different scales? Please explain.

·         Section 3.2 explains that socio-economic and demographic variables matter. How? Why not included in the conceptual model?

·         Table 5: use comma for decimals

·         Table 10: heading not complete: I guess ‘intention’ is missing.

·         Explain the models. How to they relate to the steps in the text?

·         308: 6%? How does this relate to the R2 value of 0.04?

·         Table 10 and more so 11: on my screen  line numbers are include in the table values (right column).

·         401-401: difficult to interpret. Why (not) equal?

·         454: second ‘is’ should be ‘if’.

·         461: maybe ‘…  policies THAT aim …’?

Author Response

Our responses to the Reviewer 1’s comments are in bold green

REVIEWER 1

Comments and Suggestions for Authors

I like the richness of values included, the straightforward methodology making the paper easy to understand, and the conceptual model as presented in Figure 3. But I have some concerns.

 General comments

·         There is more literature on values. The paper should make the gap(s) in the literature and the added value of this paper more explicit in the introduction.

The general literature on values is indeed large. Some general studies on values and behavior (Vinson et al., 1977, Triandis, 1979, karp, 1996, Bardi and Schwartz, 2003, Poortinga et al., 2004, Schultz, 2005), and regarding values, attitudes, intentions and behavior (Bagozzi, 1981, Maio and Olson, 1995 Vaske and Donnelly, 1999) have been added in the Introduction section However, travel behavior studies in the transport planning field that include values are very scarce. The few existing works in this field are referenced in subsections 1.2. and 1.2.

The added values of this paper is highlighted in the Introduction section: to fill the gap of contributions of travel behavior studies that include personal values as explanatory factor.

Triandis, H. C. Values, attitudes, and interpersonal behavior. En Nebraska symposium on motivation. University of Nebraska Press, 1979.

Bardi, A. and Schwartz, S.H. Values and behavior: Strength and structure of relations. Personality and social psychology bulletin, 2003, vol. 29, no 10, p. 1207-1220.

Vinson, D.E., Scott, J.E. Lamont, L. M. The role of personal values in marketing and consumer behavior. Journal of marketing, 1977, vol. 41, no 2, p. 44-50.

Karp, D. G.. Values and their effect on pro-environmental behavior. Environment and behavior, 1996, vol. 28, no 1, p. 111-133.

Poortinga, W., Steg, L., Vlek, C. Values, environmental concern, and environmental behavior: A study into household energy use. Environment and behavior, 2004, vol. 36, no 1, p. 70-93.

Schultz, P. W., et al. Values and their relationship to environmental concern and conservation behavior. Journal of cross-cultural psychology, 2005, vol. 36, no 4, p. 457-475.

Bagozzi, R. P. Attitudes, intentions, and behavior: A test of some key hypotheses. Journal of personality and social psychology, 1981, vol. 41, no 4, p. 607.

Vaske, J.J., Donnelly, M.P. A value-attitude-behavior model predicting wildland preservation voting intentions. Society & Natural Resources, 1999, vol. 12, no 6, p. 523-537.

Maio, G.R., Olson, J.M. Relations between values, attitudes, and behavioral intentions: The moderating role of attitude function. Journal of experimental social psychology, 1995, vol. 31, no 3, p. 266-285.

·         Why such a large differences (two to nine) in the number of items per value (Table 2)?

We have used the Spanish version of the Values Scale of Schwartz (Balaguer et al., 2006), which consists of 56 items, each one associated with 10 types of basic values, and distributed as shown in Table 2. The reliability and validity of the Schwartz Value Survey have been demonstrated in several works [Gouveia et al., 1998; Schwartz, 1992, 1999; Balaguer et al., 2006].

Gouveia, V.V., Clemente, M., and Vidal, M.A. (1998). El cuestionario de valores de Schwartz (CVS): propuesta de adaptación en el formato de respuesta. Revista de Psicología Social, Vol. 15, no., pp. 463-469.

Balaguer, I., Castillo, I., García-Merita, M., Guallar, A. y Pons, D. (2006). Análisis de la Estructura de Valores en los Adolescentes. Revista de Psicología General y Aplicada, 59(3), 345-357.

·         Table 3: how was intention to use asked for? And: is this about general intentions to use, or for specific cases?

To gather information regarding the intention to use each travel mode, a general question was asked to participants as follows “Which would you like to be your use of each travel mode (in percentage)?” This question collects data regarding desires. But in this study intentions and desires are very similar, because we are studying general mobility. Intentions and desires would be different if the context would have been more specific, for example in terms of time frame (Perugini y Bagozzi, 2004).

Perugini y Bagozzi. The distinction between desires and intentions, Eur. J. Soc. Psychol. 34, 69–84, 2004.

·         It seems that all modes are included in the survey. Why then the focus on cycling and walking, and not an all modes?

We focus on cycling and walking because they are the most sustainable travel modes, which is well suited for Sustainability journal. Besides, including all modes would have increased the extension of the paper up to nearly 40 pages, resulting a too long paper for a scientific journal.

·         I like figure 3, for me this is an important contribution of the paper. But if this is how it works, then why not a SEM model?

According to Schwartz (2009), SEM models are not the best method to analyze the data collected with the Schwartz Values Scale, because Exploratory Factor Analysis (EFA) “is not suitable for discovering a set of relations among variables that form a circumflex, as the values data do. The first unrotated factor represents scale use or acquiescence. It is not a substantive common factor. You can obtain a crude representation of the circular structure of values using EFA by plotting the locations of the value items on factors 2 x 3 of the unrotated solution.”

Schwartz, Shalom H. (2009). Draft Users Manual: Proper Use of the Schwarz Value Survey, version 14 January 2009, compiled by Romie F. Littrell. Auckland, New Zealand: Centre for Cross Cultural Comparisons,

·         Final section: conclusions and discussion could be distinguished more clearly.

Discussion and Conclusions have been split into two separated sections.  

A matter of taste, but it could be an option to move most tables to appendices, and condense the results in the main text.

OK

Details

·         35: ‘However, the sustainable paradigm, the recent economic recession and the new urban mobility solutions are testing this aforementioned hypothesis.’  Why?

This is just a mere hypothesis that is very difficult to check: those facts may be having an effect on changing personal values.

·         38: ‘it is well known that ….’: add references.

Bamberg, S., Rölle, D. and Weber, C. Does habitual car use not lead to more resistance to change of travel mode? Transportation, 30, 2003, 97-108.

Abrahamse, W., Steg, L., Gifford, R. and Vlek, C. Factors influencing car use for commuting and the intention to reduce it: A question of self-interest or morality? Transportation Research Part F, 12, 2009, 317–324.

Kerr, A., Lennon, A. and Watson, B. The call of the road: factors predicting students’ car travelling intentions and behavior. Transportation, 2010, 37, 1–13. DOI 10.1007/s11116-009-9217-9.

Lo, S.H., van Breukelen, G.J.P., Peters, G-J Y., and Kok, G. Commuting travel mode choice among office workers: Comparing an Extended Theory of Planned Behavior model between regions and organizational sectors. Travel Behavior and Society, 4, 2016, 1–10.

Murtagh, S., Rowe, D.A., Elliot, M.A., McMinn, D. and Nelson, N.M. Predicting active school travel: The role of planned behavior and habit strength. International Journal of Behavioral Nutrition and Physical Activity, 2012, 9:65.

Şimşekoğlu, O., Nordfjærn, T. and Rundmo, T. The role of attitudes, transport priorities, and car use habit for travel mode use and intentions to use public transportation in an urban Norwegian public. Transport Policy 42, 2015, 113–120. 

De Bruijin, G-J., Kremers, S.P.J., Schaalma, H., van Mechelen, W. and Brug, J. Determinants of adolescent bicycle use for transportation and snacking behavior. Preventive Medicine, 2005, 40, 658– 667.

Donald, I.J., Cooper, S.R. and Conchie, S.M. An extended theory of planned behaviour model of the psychological factors affecting commuters’ transport mode use. Journal of Environmental Psychology, 40, 2014, 39-48.

·         Figure 1 and related text: OK, but there are many more publications that list and cluster values. Why did you choose this one (only)?

We chose Schwartz values because is the scale most used in the very scarce literature on travel behavior and transport planning.

·         97: add dot at end sentence.

OK

·         157: May-October: why such a long period for the data collection? And sure, several people will be on holiday in August, but why then close the questionnaire? I would simply expect lower numbers of respondents, but why would there be bias?

In general, on-line surveys need long periods to increase response rates. In this way, reminders and follow-ups can be used to augment the participation. This is the only reason acknowledged in the paper to close the questionnaire on August.

·         Table 1: can you compare the sample with the population?

The distribution of the sample according to gender and education level is reasonable balanced. Considering age, those over 50 years old are under-represented in the sample. Taking into account the occupation status, participants are predominantly employed and students. We acknowledge that our study only represents students and employed people, which is a limitation of this research.

Sample

Valencia   Area

Gender

Male

48%   (596)

48%

Female

52%   (658)

52%

Age

<35  

49%   (616)

36%

35-50

30%   (374)

25%

50-70

20%   (252)

25%

>=   70

 1% (12)

14%

Occupation

Student

24%   (299)

5%

Employed

55%   (690)

25%

Others

20%   (257)

70%

Education   level

University  

53%   (668)

58%

No   university

46%   (580)

42%

·         179: why a scale ranging from MINUS 1 to seven? Why not 1-7? Which options: integers only? Why non-symmetrical? What is non-symmetrical anyway?

We have used the Spanish version of the Schwartz Values Scale (Balaguer et al., 2006), which covers 56 human values included in 10 types of basic values. Those 56 human values are measured using an equal number of items, each one associated with an asymmetric scale from -1 to 7 (-1 = opposed to my values, 0 = not important, 3 = important, 6 = very important, and 7 = of supreme importance). The asymmetry of the scale reflects distinctions that people usually make among values (Schwartz and Bardi, 2001) and, as the ranking of -1 is quite rare, there is minimal danger to the assumptions of the interval scales (Bardi et al., 2009). The reliability and validity of the Schwartz Value Scale have been demonstrated in several works [Gouveia et al., 1998; Schwartz, 1992, 1999; Balaguer et al., 2006].

Balaguer, I., Castillo, I., García-Merita, M., Guallar, A. y Pons, D.  Análisis de la Estructura de Valores en los Adolescentes. Revista de Psicología General y Aplicada, 59(3), 2006, 345-357.

Schwartz, S.H. and Bardi, A. Value hierarchies across cultures: Taking a similarities perspective. Journal of cross-cultural Psychology 32.3, 2001, 268-290.

Bardi, A., Lee, J.a., Hofmann-Towfigh, N. and Soutar, G. The structure of individual value change. Journal of personality and social psychology, 97-5, 2009, 913.

Gouveia, V.V., Clemente, M., and Vidal, M.A. El cuestionario de valores de Schwartz (CVS): propuesta de adaptación en el formato de respuesta. Revista de Psicología Social, Vol. 15, 1998, pp. 463-469.

Discussion and conclusions: Conclusions

·         Table 2 hedonism: use dot, not comma for the SD.

OK

·         179 and 200: why different scales? Please explain.

Different scales are used because we decided to use an existing and already validated values scale: Schwartz Values Scale (SVS). As explained earlier, the SVS includes a non-symmetrical scale to facilitate the understanding of the questions. On the other hand, information regarding attitudes was collected using sets of items not based in any existing scale. In these cases, we decided to use symmetrical scales because there was not need to clarify the meaning of the questions.

·         Section 3.2 explains that socio-economic and demographic variables matter. How? Why not included in the conceptual model?

We have added the following analysis in the Discussion section:

Women present significantly higher scores on Universalism, Self-direction, Hedonism, Security and Benevolence. Except for Security, these values are positively associated to walking and cycling.

Older than 40 score higher on Universalism, Security, Tradition, Conformity and Benevolence. And lower than 40 score higher on Stimulation, Hedonism and Achievement. In both cases, there is a mix of values positively and negatively associated to walking and cycling.

Those with an income higher than 1000 euro/month present higher significant scores on Power, and Security, Tradition and Conformity. And those with an income lower than 1000 euro/month (nearly half of them are students) score higher on Stimulation, Hedonism, and Achievement. The former are negatively aligned with walking and cycling, and the later are positively aligned with walking and cycling.

Students present significantly higher scores on Self-direction, Stimulation, Hedonism and Achievement, which are values positively associated with cycling and walking. On the other hand, workers present higher scores on Power and Tradition, which are values negatively associated to cycling and walking.

The conceptual model only includes psychological factors that potentially influence travel behavior.

·         Table 5: use comma for decimals

OK

·         Table 10: heading not complete: I guess ‘intention’ is missing.

The heading is correct. Former Table 10 (current Table B1) included the results of the hierarchical regression to predict the current use of bicycle. We suspect that the Reviewer is confounding Tables and their results. To improve clarity, we have included in the text references to the tables with the results of the hierarchical models.

·         Explain the models. How to they relate to the steps in the text?

Hierarchical regressions are fitted in three steps. First step only includes personal values as explanatory variables, and results are presented in Model 1. Second step includes personal values and attitudes, and results are presented in Model 2. Third step includes personal values, attitudes and intentions, and results are presented in Model 3.

·         308: 6%? How does this relate to the R2 value of 0.04?

We do not find any 6% associated to a R2 value of 0.04. We suspect that the Reviewer is confounding Tables and their results. To improve clarity, we have included references to the tables with the results of the hierarchical models.

·         Table 10 and more so 11: on my screen  line numbers are include in the table values (right column).

We have corrected this format issue.

·         401-401: difficult to interpret. Why (not) equal?

The number of significant correlations found between values and attitudes and between values and intentions is equal. And the number of significant correlations between values and behavior is lower than the previous one. This result is an indication that the association between values and attitudes and intentions is stronger than the influence of values on behavior, which confirms our conceptual framework.

·         454: second ‘is’ should be ‘if’.

OK

·         461: maybe ‘…  policies THAT aim …’?

OK

Reviewer 2 Report

Within the paper “Personal values, attitudes, and travel intentions towards cycling and walking, and actual behavior,” authors analyze the influence of personal values on attitudes, intentions, and current cycling and walking. They have identified statistically significant direct and indirect effects. The article is suitable for the journal.

Title: The title of the paper is informative. It includes important terms and the message of the article.

Abstract: The abstract describes the context and follow the structure: backgrounds, methods, results, and conclusions.

Keywords: They are well chosen. One or two should be added.

Introduction and literature review: Introduction defines the focus of the article. The literature review supports to understand the correlation of presented research results with state of the art of this research field.

Some minor remarks:

A summary table comparing the contributions of literature could support the explanation.

Line 29-30: write “[1-4]” instead of “[1, 2, 3, 4]”.

Line 102: Explain the correlation of references with the research field [28-50].

I suggest you read and add some of the articles in the special issue “Sustainable HRM” if suitable.

Materials and methods results. In my opinion, the methodology and research features are extensively discussed.

Some minor remarks:

In Table 1 the sum of percentages of occupation and education level is 99%. Please check it!

Line 179: Responses were measured on a non-symmetrical scale. How and why did you define the scale from -1 to 7?

Discussion and conclusions: Conclusions are clear and adequate. Future research directions are also discussed.

One minor remark:

Line 394, 443, 451, 455, 462, 463: check the style of citations of references!

Author Response

Our responses to the Reviewer 2’s comments are in bold green

REVIEWER 2

Comments and Suggestions for Authors

Within the paper “Personal values, attitudes, and travel intentions towards cycling and walking, and actual behavior,” authors analyze the influence of personal values on attitudes, intentions, and current cycling and walking. They have identified statistically significant direct and indirect effects. The article is suitable for the journal.

Title: The title of the paper is informative. It includes important terms and the message of the article.

Abstract: The abstract describes the context and follow the structure: backgrounds, methods, results, and conclusions.

Keywords: They are well chosen. One or two should be added.

We have added Sustainable travel

Introduction and literature review: Introduction defines the focus of the article. The literature review supports to understand the correlation of presented research results with state of the art of this research field.

Some minor remarks:

A summary table comparing the contributions of literature could support the explanation.

Table 1. Contributions of literature on travel behavior in transport planning.

Reference

Values scale

Contribution

Hunecke et al. [14]

Schwartz scale

Self-enhancement negatively   influenced bike use

Openness to change positively   affected the use of public transport

Lind et al. [15]

Value orientation measure

Awareness of consequences was   positively associated to be in the active travelers.

Ascription of responsibility was   negatively associated to be in the active travelers

Nordlund and Garvill [51]

Schwartz scale

Self-transcendence and Ecocentrism   directly influence car use reduction

Nordlund and Westin [53],

Schwartz scale

Openness to change directly influenced   on the intention to travel by a new railway line under construction.

Paulsen et al. [45],

Schwartz scale

Hedonism, Security and Power   personal values influence attitudes towards car ownership

Pojani et al [46],

Lifestyle orientations

Equality and Materialism indirectly   and positively influence the intention to use car. The former indirectly and   negatively influence the intention to use the bus. Equality was indirectly   and positively related to the intention to cycle, but Materialism directly and   negatively affected the intention to cycle.

Line 29-30: write “[1-4]” instead of “[1, 2, 3, 4]”.

OK

Line 102: Explain the correlation of references with the research field [28-50].

All those references are studies that have confirmed the existence of direct influences between intentions and real use of travel modes, or have confirmed the existence of direct influences between attitudes towards travel modes and intentions and real use of travel modes.

I suggest you read and add some of the articles in the special issue “Sustainable HRM” if suitable.

Thank you for the suggestion. We have reviewed all papers published in that special issue, and all of them are related to organization issues in companies. Unfortunately, none of them are suitable to our study.

Materials and methods results. In my opinion, the methodology and research features are extensively discussed.

Some minor remarks:

In Table 1 the sum of percentages of occupation and education level is 99%. Please check it!

OK

Line 179: Responses were measured on a non-symmetrical scale. How and why did you define the scale from -1 to 7?

We have used the Spanish version of the Schwartz Values Scale (Balaguer et al., 2006), which covers 56 human values included in 10 types of basic values. Those 56 human values are measured using an equal number of items, each one associated with an asymmetric scale from -1 to 7 (-1 = opposed to my values, 0 = not important, 3 = important, 6 = very important, and 7 = of supreme importance). The asymmetry of the scale reflects distinctions that people usually make among values (Schwartz and Bardi, 2001) and, as the ranking of -1 is quite rare, there is minimal danger to the assumptions of the interval scales (Bardi et al., 2009). The reliability and validity of the Schwartz Value Scale have been demonstrated in several works [Gouveia et al., 1998; Schwartz, 1992, 1999; Balaguer et al., 2006].

Balaguer, I., Castillo, I., García-Merita, M., Guallar, A. y Pons, D.  Análisis de la Estructura de Valores en los Adolescentes. Revista de Psicología General y Aplicada, 59(3), 2006, 345-357.

Schwartz, S.H. and Bardi, A. Value hierarchies across cultures: Taking a similarities perspective. Journal of cross-cultural Psychology 32.3, 2001, 268-290.

Bardi, A., Lee, J.a., Hofmann-Towfigh, N. and Soutar, G. The structure of individual value change. Journal of personality and social psychology, 97-5, 2009, 913.

Gouveia, V.V., Clemente, M., and Vidal, M.A. El cuestionario de valores de Schwartz (CVS): propuesta de adaptación en el formato de respuesta. Revista de Psicología Social, Vol. 15, 1998, pp. 463-469.

Discussion and conclusions: Conclusions are clear and adequate. Future research directions are also discussed.

One minor remark:

Line 394, 443, 451, 455, 462, 463: check the style of citations of references!

OK

Reviewer 3 Report

I am a transport professional with a background of engineering, economics and statistics. I am not familiar with the Schwartz Theory of Human Values, but keen on understanding the theory through reading the paper. Unfortunately, I found the paper is hard to follow with respect to the following situations:

1) The logic or justification of statistical methods that are put together deserves explanation: correlation analysis, Mann-Whitney tests and hierarchical regression. Could the analysis be carried by a more streamlined and cohesive approach such as Structural Equation Modelling (SEM), especially the paper is talking about an application of the Schwartz Theory of Human Values, which would be a perfect case for SEM? 

2) The paper did not talk about an important issue of using a convenient sample collected through the Internet: whether or not the sample is biased. See marked comments on the paper. Without the issue being properly addressed, all conclusions based on the sample may not valid as as the population in question is concerned.

3) The paper did not explain rationale of using non-symmetrical scale from -1 to 7 and 5-point Likert scales. Why are two scale regimes used?

4) For other comments, please see attached paper.

Author Response

Our responses to the Reviewer 3’s comments are in bold green

REVIEWER 3

Comments and Suggestions for Authors

I am a transport professional with a background of engineering, economics and statistics. I am not familiar with the Schwartz Theory of Human Values, but keen on understanding the theory through reading the paper. Unfortunately, I found the paper is hard to follow with respect to the following situations:

1)     The logic or justification of statistical methods that are put together deserves explanation: correlation analysis, Mann-Whitney tests and hierarchical regression. Could the analysis be carried by a more streamlined and cohesive approach such as Structural Equation Modelling (SEM), especially the paper is talking about an application of the Schwartz Theory of Human Values, which would be a perfect case for SEM?  

Pearson correlations are useful to confirm linear relationships between variables. Mann-Whitney tests are used to compare differences between two independent groups when the dependent variable is not normally distributed. And hierarchical regression is a way to show if the variables of interest explain a statistically significant amount of variance in the dependent variable after accounting for all other variables.

On the other hand, according to Schwartz (2009), SEM models are not the best method to analyze the data collected with the Schwartz Values Scale, because Exploratory Factor Analysis (EFA) “is not suitable for discovering a set of relations among variables that form a circumflex, as the values data do. The first unrotated factor represents scale use or acquiescence. It is not a substantive common factor. You can obtain a crude representation of the circular structure of values using EFA by plotting the locations of the value items on factors 2 x 3 of the unrotated solution.”

Schwartz, Shalom H. (2009). Draft Users Manual: Proper Use of the Schwarz Value Survey, version 14 January 2009, compiled by Romie F. Littrell. Auckland, New Zealand: Centre for Cross Cultural Comparisons,

2) The paper did not talk about an important issue of using a convenient sample collected through the Internet: whether or not the sample is biased. See marked comments on the paper. Without the issue being properly addressed, all conclusions based on the sample may not valid as as the population in question is concerned.

The study focuses on the Valencia region. We have included statistics for the region in Table 1. The distribution of the sample according to gender and education level is reasonable balanced. Considering age, those over 50 years old are under-represented in the sample. Taking into account the occupation status, participants are predominantly employed and students. We acknowledge that our study only represents students and employed people, which is a limitation of this research. In this case, Anova analysis method is not applicable because variance information from the population is not available.

The paper did not explain rationale of using non-symmetrical scale from -1 to 7 and 5-point Likert scales. Why are two scale regimes used?

We have used the Spanish version of the Schwartz Values Scale (Balaguer et al., 2006), which covers 56 human values included in 10 types of basic values. Those 56 human values are measured using an equal number of items, each one associated with an asymmetric scale from -1 to 7 (-1 = opposed to my values, 0 = not important, 3 = important, 6 = very important, and 7 = of supreme importance). The asymmetry of the scale reflects distinctions that people usually make among values (Schwartz and Bardi, 2001) and, as the ranking of -1 is quite rare, there is minimal danger to the assumptions of the interval scales (Bardi et al., 2009). The reliability and validity of the Schwartz Value Scale have been demonstrated in several works [Gouveia et al., 1998; Schwartz, 1992, 1999; Balaguer et al., 2006]. On the other hand, information regarding attitudes was collected using sets of items not based in any existing scale. In these cases, we decided to use symmetrical scales because there was not need to clarify the meaning of the questions.

Balaguer, I., Castillo, I., García-Merita, M., Guallar, A. y Pons, D.  Análisis de la Estructura de Valores en los Adolescentes. Revista de Psicología General y Aplicada, 59(3), 2006, 345-357.

Schwartz, S.H. and Bardi, A. Value hierarchies across cultures: Taking a similarities perspective. Journal of cross-cultural Psychology 32.3, 2001, 268-290.

Bardi, A., Lee, J.a., Hofmann-Towfigh, N. and Soutar, G. The structure of individual value change. Journal of personality and social psychology, 97-5, 2009, 913.

Gouveia, V.V., Clemente, M., and Vidal, M.A. El cuestionario de valores de Schwartz (CVS): propuesta de adaptación en el formato de respuesta. Revista de Psicología Social, Vol. 15, 1998, pp. 463-469.

2)     For other comments, please see attached paper.

OK

Round 2

Reviewer 1 Report

The authors have clearly responded to my comments. I personally would explain to the reader why a SEM model could not be estimated, but that is a matter of taste.

Reviewer 3 Report

This is a significant improved version of the paper. Although the paper has room for further improvement as far my comments are concerned, I am pleased with with current revision.